# KNOTHE-ROSENBLATT TRANSPORT FOR UNSUPERVISED DOMAIN ADAPTATION

## ABSTRACT

Unsupervised domain adaptation (UDA) aims at exploiting related but different data sources to tackle a common task in a target domain. UDA remains a central yet challenging problem in machine learning. In this paper, we present an approach tailored to moderate-dimensional tabular problems which are hugely important in industrial applications and less well-served by the plethora of methods designed for image and language data. Knothe-Rosenblatt Domain Adaptation (KRDA) is based on the Knothe-Rosenblatt transport: we exploit autoregressive density estimation algorithms to accurately model the different sources by an autoregressive model using a mixture of Gaussians. KRDA then takes advantage of the triangularity of the autoregressive models to build an explicit mapping of the source samples into the target domain. We show that the transfer map built by KRDA preserves each component quantiles of the observations, hence aligning the representations of the different data sets in the same target domain. Finally, we show that KRDA has state-of-the-art performance on both synthetic and real world UDA problems.

## 1 INTRODUCTION

In classical machine learning, we assume that both the training and test data follow the same distribution and we can thus expect to generalize from the training set to the test set. In practice, this assumption does not always hold. For example, data is often collected in asynchronous manner, at different times and locations, and may be labeled by different people, which can affect the efficiency and quality of the standard supervised learning models (Quionero-Candela et al., 2009; Pan & Yang, 2010). Collecting data from multiple sources may also lead to distribution shift between the collectors. For example, wireless network data would present different properties and patterns depending on time (such as day, night, week-end), or location/ infrastructure (downtown, countryside, or touristic area). Even when the task is common, an efficient approach should take into account the shift. Coping with this problem lead to the development of *transfer learning* methods that adapt the knowledge from a source domain to a new target domain.

Transfer learning, or domain adaptation, is central in vision (image classification, image segmentation, or activity recognition) (Li et al., 2020b) and natural language processing (translation, language generation) (Malte & Ratadiya, 2019; Ruder et al., 2019) problems. Both of these domains generate very high-dimensional data, and transfer learning usually focuses on fine tuning pre-trained models to specific tasks. In contrast, the principal problem in many industrial applications is not dimensionality, rather class imbalance, probability shift in data collection, and small data (Zhang et al., 2019). For example, wireless network data (5G and beyond), IoT or smart cities are often low dimensional (less than 100), and highly dependent on the data collection context (Fu et al., 2018; Arjoune & Faruque, 2020; Benzaid & Taleb, 2020). These issues are rarely dealt within the transfer learning literature.

Transfer learning on high-dimensional data usually proceeds by mapping the data into a smaller dimensional space and carrying out the transfer in this latent space. In the lower dimensional domain we are targeting, we can use recently developed powerful density estimation techniques and principled transport-based approaches that rely on these precise estimates.

Our contribution is *Knothe-Rosenblatt Domain Adaptation*, or KRDA. We tackle *Domain Adaptation* (DA) which arises when the probability distribution of the source and the target data are different but related. We focus on the more challenging task where we do not have labeled target data. This approach, called *Unsupervised Domain Adaptation*, is the most difficult case of distribution shift.

We estimate the density of both the source and the target data in order to transfer the former to the later. We use RNADE (Uria et al., 2013), an autoregressive technique that decomposes the $d$-dimensional density into $d$ one-dimensional conditional densities, represented by input dependent mixtures of Gaussian (also known as mixture density nets (Bishop, 1994)). Using these explicit representations, KRDA transfers each sample by preserving the conditional quantiles with Knothe-Rosenblatt transport. Once embedded in the target domain, the source and its labels are learned by a supervised learning algorithm. Although theoretically simple, using autoregressive models in order to perform a transport has not yet been considered in the transfer learning literature.

As it will be illustrated, KRDA is particularly well suited for small data (less than 10000 samples) in small dimension (less than 100), where other state-of-the-art methods tend to under-perform, as shown in Section 6. We can also consider KRDA as an embedding algorithm with a great advantage: all the extra-computational cost of KRDA is spent in the computation of the transfer map. Once the source is transferred, training and testing will have no overhead beyond the cost of the supervised learning algorithm used.

The paper is structured as follows. We first introduce KRDA, an algorithm based on density estimation. We review some topics in density estimation in Section 4. We then introduce KRDA, the core of our paper, in Section 5, and expose its properties and limitations. Finally we compare our approach against state-of-the-art transfer learning algorithms on several benchmark. A detailed experimental setting and the results are given in Section 6.

## 2 RELATED WORK

Transfer learning (TL) aims at building algorithms that generalize across different domains with different probability distributions, see for example (Pan & Yang, 2010; Kouw & Loog, 2019; Zhuang et al., 2019) for global surveys of the field. *Domain adaptation* is the specific case when the task is the same across the different domains. DA approaches may be roughly divided in two categories depending on whether we have access to labels in the target space, or not. The first case is known as *semi-supervised DA*. The usual approach is to find a global transformation that aligns the different domains by preserving the information coming from a few labels (Saenko et al., 2010). Many papers embed both domains in the same latent space using different tools such as similarity (Donahue et al., 2013), non-linear kernel mapping (Pan et al., 2010; Gong et al., 2016), or entropy (Saito et al., 2019).

In *unsupervised domain adaptation*, we assume that we have no labels from the target domain. One avenue is to reweight the samples in order to correct the shift between the source and target distributions (Huang et al., 2007; Gretton et al., 2007). This method has the advantage of not requiring distribution estimation or specific embedding. As in semi-supervised DA, other unsupervised approaches rely on a common latent space. Both the source and target data are projected into this space, and a classifier is then learned using the labeled source data in the latent space. Another shallow approach, subspace mapping, aims at learning a linear map that aligns source and target (Gong et al., 2012; Fernando et al., 2013; Sharma et al., 2012).

More recently, deep neural nets became a popular choice in UDA due to the flexibility of these models to learn rich non-linear mappings. Deep Adaptation Network (Long et al., 2015) adds multiple kernel variants of MMD at the top layers to push the target distribution close to the source. Domain-Adversarial Neural Network (Ganin et al., 2016) introduces adversarial training to reduce the distance between the source and target feature distributions. Joint Adaptation Networks (Long et al., 2017) and Conditional Adversarial Domain Adaptation (Long et al., 2018) aim at aligning the joint or conditional distributions. Instead of learning transferable representations, Saito et al. (2018) align the source and target distributions by maximizing the discrepancy between the outputs of two classifiers. Using clustering is another approach (Shu et al., 2018; Liang et al., 2020; Li et al., 2020a).

In this paper we use recently developed powerful density estimators to relate the source and target domains. Density estimation is an important problem in statistics in general and machine learning in particular (Bishop, 1994; Wasserman, 2004). Among the plethora of methods (Salakhutdinov & Hinton, 2009; 458; Rezende & Mohamed, 2015; Ho et al., 2019), we use autoregressive models for their triangularity that is crucial for our approach (Larochelle & Murray, 2011; Uria et al., 2013). These algorithms model the joint distribution as product of one-dimensional conditional densities using the probability chain rule. They take advantage of recent developments in recurrent neural

networks (Oord et al., 2016). A drawback of this approach is the fixed arbitrary ordering of the components, although it seems not to be crucial in many applications (Kégl et al., 2021), including ours, arguably explained by the flexibility of the mixtures that can model the potentially complex conditional densities.

Optimal transport sees the domain adaptation problem as graph matching (Courty et al., 2017) and embed the source into the target by minimizing a transportation cost. Knothe-Rosenblatt transport has been independently introduced in (Rosenblatt, 1952; Knothe, 1957), the former for multivariate statistics analysis and the latter to study isoperimetric inequality problems. This approach has been applied for histogram equalization for RGB pictures (Pitié et al., 2007). More recently, (Muzellec & Cuturi, 2019) use a generalization of Knothe-Rosenblatt transport as a surrogate to optimal transport in high dimensional spaces: the paper fits a multi-dimensional GMM that is transferred at once.

## 3 BACKGROUND AND NOTATIONS

We consider the setting of classical transfer learning. The fundamental objective is to use some knowledge acquired during the learning of a specific predictive task in order to perform a similar but different task. More precisely, the *domain $\mathcal{D}$* of a learning task is a couple composed of a *feature space $\mathcal{X}$* and a marginal probability distribution $p(X)$. A *task* is a couple $(\mathcal{Y}, f)$ composed of a label space $\mathcal{Y}$ and a prediction function $f : \mathcal{X} \to \mathcal{Y}$. In transfer learning, we consider two domains and learning tasks named *source* $(\mathcal{D}_\mathcal{S}, \mathcal{Y}_\mathcal{S})$ and *target* $(\mathcal{D}_\mathcal{T}, \mathcal{Y}_\mathcal{T})$. In the transfer learning setting $(\mathcal{X}_\mathcal{S}, p_\mathcal{S}) \neq (\mathcal{X}_\mathcal{T}, p_\mathcal{T})$ and the goal is to transfer some knowledge from the source to the target. There are several sub-cases such as *covariate shift*, on which KRDA relies, in which $p_\mathcal{S}(x) \neq p_\mathcal{T}(x)$ but the conditional probabilities are invariant: $p_\mathcal{S}(y|x) = p_\mathcal{T}(y|x)$ for every $y \in \mathcal{Y}$.

In the *unsupervised* setting we have no access to the labels of the target data. We thus aim at building a transfer map $T : \mathcal{X}_\mathcal{S} \to \mathcal{X}_\mathcal{T}$ which associates a vector in the target feature space to every source sample before applying a classifier.

Let $D_\mathcal{S} = (X_\mathcal{S}, Y_\mathcal{S})$ and $D_\mathcal{T} = (X_\mathcal{T}, Y_\mathcal{T})$ be the source and target data sets, respectively. Let $p_\mathcal{S}$ and $p_\mathcal{T}$ be the probability density functions (PDF) of the source data $X_\mathcal{S}$ and target data $X_\mathcal{T}$, and let $\hat{p}_\mathcal{S}$ and $\hat{p}_\mathcal{T}$ be the estimated densities, respectively. We will denote by $F_p$ the cumulative density function (CDF) associated with the density $p$. For a vector function $g : \mathbb{R}^m \to \mathbb{R}^n$ and $x \in \mathbb{R}^m$, let $g^i(x) \in \mathbb{R}$ be the $i$-th coefficient of $g(x)$.

## 4 AUTOREGRESSIVE DENSITY ESTIMATION

In this work, we will focus on autoregressive models. The probability density function is expressed using the *probability chain rule*: the PDF of a vector $x = (x^1, \ldots, x^d) \in \mathbb{R}^d$ is the product of one-dimensional conditional densities

$$p(x) = \prod_{i=1}^{d} p^i(x^i | x^{<i}). \tag{1}$$

Each conditional factor density will be approximated by a Gaussian mixture distribution. Note that this straightforwardly generalizes to any type of mixture distribution, although we will focus on Gaussian mixtures in this paper for didactic purposes.

RNADE (Uria et al., 2013) is a robust and flexible deep learning method that, following Eq. (1), fits one-dimensional conditional Gaussian mixtures (originally proposed by (Bishop, 1994) under the name of mixture density net (MDN)) for every coefficient of a vector $x = (x^1, \ldots, x^d) \in \mathbb{R}^d$. More precisely we associate to each conditional probability $p^i(x^i | x^{<i})$ a distribution composed of a mixture of $N$ Gaussians $\sum_{k=1}^{N} w_k^i \mathcal{N}(\mu_k^i, \sigma_k^i)$. The RNADE algorithm with hidden size $H$ is based on NADE (Larochelle & Murray, 2011) and can be summarized as follows. We first compute from the input $x = (x^1, \ldots, x^d)$ the sequence $a^i \in \mathbb{R}^H$, $i = 1, \ldots, d$, in an iterative manner:

$$a^1 = c; \qquad\qquad a^{i+1} = a^i + x^i W_{\cdot, i}, \tag{2}$$

where $c \in \mathbb{R}^H$ and $W \in \mathbb{R}^{H \times d}$ are learned parameters, and $W_{\cdot, i}$ denotes the $i$th column of the parameter matrix $W$. We then apply a non-linearity after re-scaling

$$h^i = \sigma(C^i a^i) \tag{3}$$

to get the parameters of the conditional Gaussian mixture as output of linear layers:

$$w^i = \text{Softmax}(\text{Lin}_1(h^i)), \tag{4}$$

$$\mu^i = \text{Lin}_2(h^i), \tag{5}$$

$$\sigma^i = \exp(0.5 \times \text{Lin}_3(h^i)), \tag{6}$$

where $\text{Lin}_1, \text{Lin}_2, \text{Lin}_3$ are three linear layers $\mathbb{R}^H \to \mathbb{R}^N$ with bias. In this work, we use $\sigma = \text{RELU}$ as the non-linearity applied in (3). The exact likelihood is thus directly accessible, and the model is trained end-to-end by maximizing the log-likelihood using gradient ascent. Note that this density estimator also makes data generation from the estimated distribution easy (go through the chain Eq. (1) and sample from Gaussian mixtures).

In domain adaptation, we make the assumption of having related distributions for the source and the target data. In order for the density estimation model to use this assumption, we share the parameters $c$ and $W$ in Eq. (2) for all data sets. The last linear layer of Eqs. (4-6) are specific to the source and target and will capture the dissimilarities between the domains. The density estimation network is then trained simultaneously on both the source and the target data.

## 5 KNOTHE-ROSENBLATT TRANSPORT

Having two densities $p_\mu$ and $p_\nu$, there are several ways to built a *transport* place $T$ such that $T_\sharp \mu = \nu$. For example, the change of variable formula $p_\mu = p_\nu(T(x)) \det(Jac_x T)$ defines a PDE for which $T$ is solution (assuming existence). However this direct approach is not tractable in general. We propose here to use our autoregressive density estimation model in order to build a Knothe-Rosenblatt transport map. We refer to (Villani, 2008; Santambrogio, 2015) for an extensive presentation and study on Optimal Transport (OT) in general and Knothe-Rosenblatt (KR) transport in particular.

### 5.1 KNOTHE-ROSENBLATT TRANSPORT

Let $\mu$ and $\nu$ two absolutely continuous measures of $\mathbb{R}$ with $F(x) = \int_{-\infty}^x d\mu$ and $G(x) = \int_{-\infty}^x d\nu$ their cumulative distribution function (CDF). We define the pseudo inverse of the CDF $F$ as

$$F^{-1}(x) = \inf\{z \in \mathbb{R} : F(z) > x\}.$$

The following theorem gives a transportation map (actually optimal) between $\mu$ and $\nu$.

**Theorem 1** ((Santambrogio, 2015, Theorem 2.5)). *The map $T = G^{-1} \circ F$ verifies $T_\# \mu = \nu$.*

Knothe-Rosenblatt transport (Rosenblatt, 1952; Knothe, 1957) is a simple transportation plan that applies one-dimensional optimal transport to all conditional marginals of one distribution into another. For didactic purposes, we give here a definition involving only density functions defined through the Lebesgue measure, *i.e.* we write $\mu(A) = \int_A f dx$ where $f$ is the density of the probability of $\mu$. Let $p_S$ and $p_T$ be two density functions on $\mathbb{R}^d$, hence $p_S = \mu_S \, d\lambda(\mathbb{R}^d)$ and $p_T = \nu_T \, d\lambda(\mathbb{R}^d)$.

Consider the first marginals $p_S(x_1)$ and $p_T(x_1)$ as one-dimensional random variables. By Theorem 1, we have a transport map $T_1 : \mathbb{R}^1 \to \mathbb{R}$ such that for $x_1 \sim p_S(x_1)$, we have $T_1(x_1) \sim p_T(x_1)$. Now consider the conditional marginal $p_S(x_2|x_1)$ and $p_T(x_2|x_1)$, by Theorem 1 we construct again a map $T_2 : \mathbb{R}^2 \to \mathbb{R}$ such that for $x_1 \sim p_S(x_1)$ and $x_2 \sim p_S(x_2|x_1)$, we have $T_2(x_1, x_2) \sim p_T(x_2|x_1)$. By iterating the previous process for all components, we construct a collection of $d$ maps $T_1, \dots, T_d$. The Knothe-Rosenblatt transport is the map that sends $x \in \mathbb{R}^d$ to $\mathbb{R}^d$ by applying this construction to all conditional marginals in the following way:

$$T(x_1, \dots, x_d) = (T_1(x_1), T_2(x_1, x_2), \dots, T_d(x_1, \dots, x_d)). \tag{7}$$

The following theorem assures the correctness of this approach as the density of the source is perfectly mapped on the target in the following sense.

**Proposition 2** ((Santambrogio, 2015, Proposition 2.18)). *The map $T$ satisfies $T_\# \mu_S = \nu_T$.*

**Relationship with Optimal Transport** In one dimension, KR transport and OT coincide. Hence, KR transport optimally couples all conditionals. More generally Carlier et al. (2009) show that KR is a limit of optimal transport with quadratic costs $l_\lambda(x, y) = \sum_i \lambda_i(x_i - y_i)$ when $\lambda_i/\lambda_{i+1} \to 0$. By proceeding coefficient after coefficient instead of globally such as OT, KR transport might offer some interesting regularization for the specific case of transfer learning.

## 5.2 KNOTHE-ROSENBLATT DOMAIN ADAPTATION

In this section, we present the KRDA algorithm (for *Knothe-Rosenblatt Domain Adaptation*), the main contribution of the paper. We follow the Knothe-Rosenblatt construction of a transportation map based on estimated densities of the source and the target domain. In order to maximally exploit the conditional marginal structure of the map, we are relying on autoregressive density estimation models such as RNADE.

Let $X_{\mathcal{S}}$ the source input data set with associated density $p_{\mathcal{S}}$. Modeling the PDF as a conditional mixture of Gaussians for every component, we obtain $\hat{p}_{\mathcal{S}}$. For $x \in X_{\mathcal{S}}$ we estimate $\hat{p}_{\mathcal{S}}^i(x^i) = \sum_k w_k^i \mathcal{N}(\mu_k^i, \sigma_k^i)$. The CDF $F$ is easily obtained by linearity from the PDF as

$$F^i(x^i) = \sum_k w_k^i F_{\mathcal{N}(\mu_k^i, \sigma_k^i)}(x^i), \tag{8}$$

where $F_{\mathcal{N}(\mu^i, \sigma^i)}$ is the CDF of the Gaussian $\mathcal{N}(\mu^i, \sigma^i)$. KRDA builds a transfer function $T : \mathcal{X}_{\mathcal{S}} \to \mathcal{X}_{\mathcal{T}}$ such that for every sample $x \in \mathcal{X}_{\mathcal{S}}$, we have $\hat{F}_{\mathcal{S}}^i(x^i) = \hat{F}_{\mathcal{T}}^i(T(x^i))$, or $T(x^i) = \hat{F}_{\mathcal{T}}^{i-1} \circ \hat{F}_{\mathcal{S}}^i(x^i)$ ($\hat{F}_{\mathcal{T}}^{i-1}$ is the generalized inverse distribution function as in Equation 7). Theorem 1 assures that taking $T = \hat{F}_{\mathcal{T}}^{i-1} \circ \hat{F}_{\mathcal{S}}^i$ for every scalar component $x^i$ maintains the previous property.

KRDA relies on Proposition 2 in the following sense. With RNADE, we first estimate the densities $\hat{p}_{\mathcal{S}}$ and $\hat{p}_{\mathcal{T}}$ of the source and the target data, before transferring all samples from the data set $X_{\mathcal{S}}$ to the target domain $\mathcal{X}_{\mathcal{T}}$. Let $x_{\mathcal{S}} = (x^1, \ldots, x^d) \in X_{\mathcal{S}}$. From $\hat{p}^i(x_{\mathcal{S}}^i)$, we compute $\hat{F}^i(x_{\mathcal{S}}^i)$ and using Proposition 2 we compute $T(x^i) \in \mathcal{X}_{\mathcal{T}}$ such that $\hat{F}_{\mathcal{S}}^i(x^i) = \hat{F}_{\mathcal{T}}^i(T(x^i))$. Note that autoregressive models are triangular in the sense that $F^i(x^i)$ does only depend on coefficients $x^{<i} = (x^j)^{j<i}$. After having estimated the density of both the source and the target data, we thus construct $T(x)$ deterministically, component by component.

1. $p_{\mathcal{T}}^1(x_{\mathcal{T}}^1)$ is fully determined by $w^1$, $\mu^1$ and $\sigma^1$ that depend only on the parameter $c$, and hence is independent of $x_{\mathcal{S}}^1$. We then assign $x_{\mathcal{T}}^1 = \hat{F}_{\mathcal{T}}^{1-1} \circ \hat{F}_{\mathcal{S}}^1(x_{\mathcal{S}}^1)$ so that we have $\hat{F}_{\mathcal{T}}^1(x_{\mathcal{T}}^1) = \hat{F}_{\mathcal{S}}^1(x_{\mathcal{S}}^1)$.

2. $p_{\mathcal{T}}^i(x_{\mathcal{T}}^i | x_{\mathcal{T}}^{\leq i})$ (or the neural net outputs $w^i$, $\mu^i$ and $\sigma^i$) only depends on the components $x_{\mathcal{T}}^{\leq i}$. We then run RNADE on the partially computed $x_{\mathcal{T}}$ in order to access the Gaussian mixture. As previously, we assign $x_{\mathcal{T}}^i = \hat{F}_{\mathcal{T}}^{i-1} \circ \hat{F}_{\mathcal{S}}^i(x_{\mathcal{S}}^i)$ and we keep the quantile invariant property by construction.

This procedure terminates once all components of $x_{\mathcal{T}}$ have been computed, hence after $d$ steps in order to obtain $x_{\mathcal{T}} \in \mathcal{X}_{\mathcal{T}}$ such that

$$\forall i \in \{1, \ldots, d\}, \ \hat{F}^i(x_{\mathcal{S}}^i) = \hat{F}^i(x_{\mathcal{T}}^i). \tag{9}$$

Note that in the case where the source and target domains are the same, *i.e.* $p_{\mathcal{S}} = p_{\mathcal{T}}$, we have $T = \mathrm{id}$.

At every step, the computation of the inverse of the CDF is done with a binary search: the CDF is monotonically increasing and $F^{-1}(x)$ is a zero of the function $z \mapsto F(z) - x$. The complete description of the algorithm is given in Algorithm 1, and we provide several visualizations of the transfer map in Section 6.

Once the whole source data $X_{\mathcal{S}}$ is transferred into the target domain $T(X_{\mathcal{S}}) \subset \mathcal{X}_{\mathcal{T}}$, a supervised learning algorithm (e.g., an SVM) is trained on the (labelled) data composed of the transferred source samples with their labels $(T(X_{\mathcal{S}}), Y_{\mathcal{S}})$. At test time, samples from the target domain are directly given to the learning algorithms for prediction, hence giving no overhead cost after the transfer phase.

**Implementation details** We perform the CDF inversion by exploiting its non-decreasing property. To compute $F^{-1}(x)$ we use a bisection algorithm to find a zero of the function $z \mapsto F(z) - x$. The initial search interval is iteratively determined by the presence of a root in $[-2^k, 2^k]$ for growing $k$. For numerical stability, we clip the values of $x$ to $[\epsilon, 1 - \epsilon]$, with $\epsilon = 10^{-8}$. The transfer algorithm is parallelizable since each sample can be treated independently.

---

**Algorithm 1** KRDA

    Learn $\hat{p}_{\mathcal{S}}$ on $X_{\mathcal{S}}$
    Learn $\hat{p}_{\mathcal{T}}$ on $X_{\mathcal{T}}$
    **for** $x_{\mathcal{S}} \in \mathcal{S}$ **do**
        Initialize $x_{\mathcal{T}} = 0 \in \mathbb{R}^d$
        **for** $i \in \{1, \ldots, d\}$ **do**
            Compute $\hat{F}^i(x_{\mathcal{S}}^i)$
            Compute the partial CDF $\hat{F}_{\mathcal{T}}^{i\,-1}$ from $\hat{p}_{\mathcal{T}}^i$
            Compute $x_{\mathcal{T}}^i = \hat{F}_{\mathcal{T}}^{i\,-1} \circ \hat{F}_{\mathcal{S}}^i(x_{\mathcal{S}}^i)$
        **end for**
        Set $T(x_{\mathcal{S}}) \leftarrow x_{\mathcal{T}}$
    **end for**
    ▷ run learning algorithm (e.g. SVM) on $\big(T(X_{\mathcal{S}}), Y_{\mathcal{S}}\big)$

---

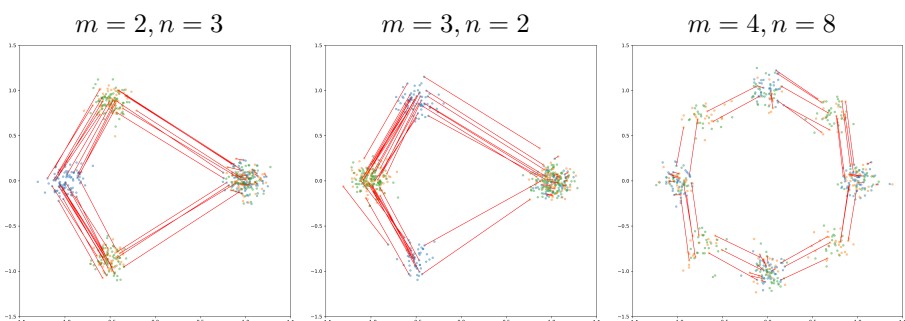

Figure 1: $m$-Gaussian mixture transferred with KRDA to an $n$-Gaussian mixture; we plot the source (blue), target (green), transferred (orange) and some mappings (red).

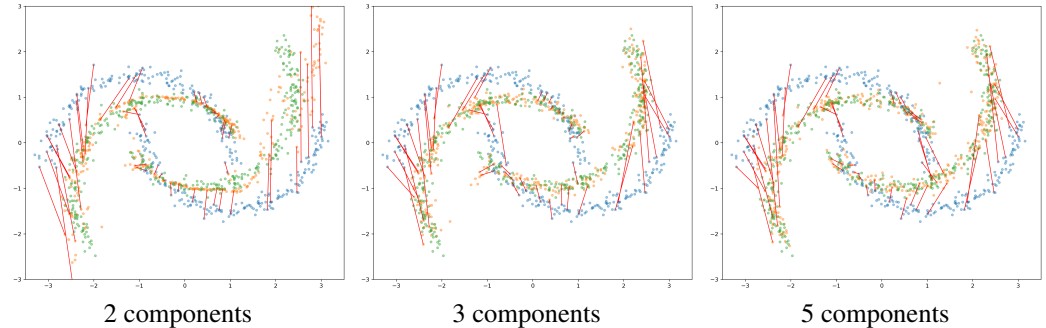

Figure 2: KRDA on the inter-twinning moon dataset with different number of components; we plot the source (blue), target (green), transferred (orange) and some mappings (red).

**Limitations**    KRDA has two main limitations. The first one is the reliance of KRDA on the estimation made by RNADE. Since RNADE does not generalize well to high-dimensional spaces ($\geq 100$), we cannot expect KRDA to work well in this regime. The second limitation is the computation cost of the transfer: the algorithm iterates on the dataset (parallelizable) and on the components (not parallelizable because of the conditional distributions). Even though KRDA is linear in the number of samples, it may not be suited for very large datasets of millions of samples. As we do not use label information, our approach is suited for the covariate-shift setting.

## 6   Experiments

### 6.1   Domain adaptation

In this section, we evaluate KRDA in both synthetic and real data sets. In all KRDA experiments, we model the source and target distributions with mixtures of five Gaussian components. All the

following experiments are performed in the unsupervised domain adaptation setting, when no labels of the target domain are accessible. After transformation, we apply an SVM on the transferred source. We compare our method to various techniques which can be classified into three classes:

- Baseline solutions: For a specific ML model, here Support Vector Machine (SVM), **Source only** learns this model using source data set with its labels; **Target only** learns the model by using the labeled target data (note that none of the transfer competitors have access the target data set labels, so this is meant as an optimistic baseline);
- Shallow solutions: Subspace Alignment **SA**, (Fernando et al., 2013) and reweighting methods Transfer Component Analysis **TCA** (Pan et al., 2011), Kernel Mean Matching **KMM**(Gretton et al., 2009) , each followed by a SVM;
- Optimal Transport **OT** (Courty et al., 2017), followed by a **SVM**;
- Deep learning solutions include **DAN** (Long et al., 2015), **DANN** (Ganin et al., 2016), **JAN** (Long et al., 2017), **CDAN** (Long et al., 2018) and **SHOT** (Liang et al., 2020). We implement DAN, DANN, JAN and CDAN using the `dalib` library (Junguang Jiang, 2020). Instead of using the original image classifier in `dalib` that would be unsuitable in our benchmark, we use a five-layer MLP for each algorithm. For SHOT, the feature net is a four-layer MLP and the classifier net is a one-layer regression model a total of five layers.

For space reasons, we only show a sub-selection of the benchmark: the complete tables are located in Appendix A.

**Hyperparameter Tuning**   We fix most hyperparameters in our experiments. In deep learning models (DANN, JAN, and CDAN) $\eta = 1$. The number of hidden neurons of MLP is set to 250 which will be equivalent to the number of parameters of the KRDA models. All the deep learning models are trained using the Adam optimizer (Kingma & Ba, 2015) with a learning rate set to $10^{-3}$.

**Metric and cross validation**   Accuracy is used as the metric to evaluate the performance of different algorithms in the following experiments. All experiments have two classes. We run each experiment five times. In each running, we randomly pick 90% source and 90% target data to train the model. The test is systematically performed on unseen samples from the target data set. In each table, we report the average accuracy of the five experiments as well as 95% confidence intervals. The code used in all our experiments is available on Github[1].

## 6.2 SYNTHETIC DATA EXPERIMENTS

**Mixtures of Gaussians**   We generate a mixture of Gaussians for the source and target data. We use 1000 samples in both data sets, and we are interested in seeing how the domain adaptation tasks are handled by KRDA. For all tasks, RNADE uses $N = 5$ Gaussian components and a hidden layer of dimension 50. We plot several transfers from mixtures of $m$ Gaussians to mixtures of $n$ Gaussians. The results are presented in Figure 1.

**Inter-twinning moons**   We perform three experiments on the classical inter-twinning moons dataset. In all these experiments, KRDA uses $N = 5$ Gaussian components and a hidden layer of dimension 50. The inter-twinning moon dataset is composed of two interlacing half-moons with labels 0 and 1.

1. We show the KRDA embedding from the source to a target domain with different components of the Gaussian mixture. We highlight the transfer of the same set of source sample in all figures. Both train and target data are of size 1000, the target distribution is a $40°$ rotation of the source distribution. These visualizations are shown in Figure 2.

2. We use the same experimental setup as in (Germain et al., 2013). We sample 300 samples from the source distribution and 300 in the target domain with various angles between $10°$ and $90°$. The difficulty of the problem increases with the angle. The test set is composed of 1000 samples from the target distribution. We show the performance of KRDA and competitors in Table 4.

3. KRDA is run in six inter-twining moons tasks to investigate its performance in different training sizes. The source and target training data size ranges from 200 to $1,000$. In each task, the target data

---

[1]It will be released after the review process.

distribution is rotated with $40°$ from the source. Following previous cross validation setting, in each task we run each algorithm five times with randomly picked 90% source and 90% target data and average the results. Figure 3 shows results and the corresponding 95% confidence interval. We find that although all algorithms tend to converge, KRDA acts excellently in small size cases and it is stabler than other algorithms in the low-data setting.

Table 1: Inter-twinning moons unsupervised

| Task | Source | Target | DANN | SA | KMM | OT | SHOT | KRDA |
|---|---|---|---|---|---|---|---|---|
| $10°$ | 100±0.00 | 100±0 | **100±0.00** | 85.3±0.5 | 50.1±12.0 | **100.0±0.0** | 82.8±10.8 | **100±0.0** |
| $20°$ | 99.9±0.1 | 100±0 | 99.3±0.5 | 78.5±0.4 | 53.0±8.8 | **100.0±0.0** | 81.4±3.7 | **100±0.0** |
| $30°$ | 96.6±0.4 | 100±0 | 89.4±7.2 | 73.4±0.3 | 51.4±10.9 | **99.8±0.2** | 77.1±2.1 | **100±0.0** |
| $40°$ | 73.1±2.5 | 100±0 | 73.8±21.3 | 69.2±0.2 | 53.4±27.3 | 89.6±1.2 | 24.4±2.2 | **98.4±2.2** |
| $50°$ | 41.5±2.0 | 100±0 | 48.5±21.9 | 61.8±0.5 | 56.3±7.8 | 83.8±0.8 | 22.3±1.6 | **98.4±1.5** |
| $60°$ | 28.7±0.6 | 100±0 | 44.2±14.3 | 54.8±0.4 | 47.8±18.1 | 78.0±1.3 | 19.9±1.3 | **90.5±2.8** |
| $70°$ | 23.3±0.4 | 100±0 | 24.3±1.6 | 49.0±0.4 | 52.5±9.0 | 71.6±0.6 | 17.4±1.2 | **84.4±0.8** |
| $80°$ | 20.4±1.70 | 100±0 | 20.1±2.4 | 43.2±0.6 | 48.2±9.1 | 65.8±0.7 | 15.7±1.3 | **81.2±3.4** |
| $90°$ | 18.1±0.3 | 100±0 | 17.7±1.5 | 38.2±0.1 | 54.4±17.9 | 59.4±1.2 | 14.8±1.3 | **71.1±3.9** |

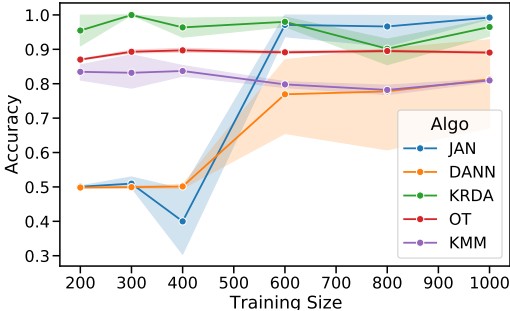

Figure 3: The learning curves of different UDA algorithms under different source and target training sizes. KRDA has excellent performance on small data sets and it shows very stable accuracy over different training sizes.

## 6.3 EXPERIMENTS ON REAL DATA

**Hepmass** The HEPMASS data comes from high-energy physics (Baldi et al., 2016). The objective of the task is to learn how to separate exotic particles from background noise using collision data (27 measured features). The data may be split according to the mass of the observed particles ($m \in \{500, 750, 1000, 1250, 1500\}$). We create different transfer learning tasks: transferring the domain from one mass to another. We build the source and target data by subsampling 1000 and 500 instances from the original data with a given source and target mass, respectively. We also sample 2000 independent instances from the target domain for the test set. KRDA uses $N = 5$ Gaussian components and dimension 100 for its hidden layer. Each transfer of the source data is followed by a SVM with the hyperparameters shared for all experiments that require it. SA uses 10 components. The results are summarized in Table 5. As shown, KRDA performs similarly and often better than state-of-the-art competitors.

**Amazon dataset** This data (McAuley et al., 2015) is an aggregation of reviews from four different products (dvd (D), books (B), electronics (E), kitchen (K)) and their given grade by customers. Each product or domain has about 2000 training samples and 4000 test samples. Each sample is presented by 5000 features and is associated to a binary class: 0 for samples ranked less than three stars and 1 otherwise. The goal is to transfer the review-to-grade classification from one product to another. Thus we created twelve transfer tasks using these products. As KRDA is based on RNADE for the density estimation, and RNADE is not designed for high dimensional data, we used a neural net (NN)

Table 2: HEPMASS unsupervised domain adaptation

| Task | Source | Target | DANN | SA | KMM | OT | SHOT | KRDA |
|---|---|---|---|---|---|---|---|---|
| 500 →750 | 68.7±0.1 | 82.2±0.3 | 56.7±2.9 | 67.9±1.2 | 60.7±3.5 | 67.6±2.0 | **74.1±1.9** | **75.6±2.3** |
| 500 →1000 | 67.2±0.1 | 89.2±0.4 | 56.7±4.4 | 71.7±1.0 | 50.1±10.8 | 72.1±1.9 | **85.5±1.1** | 80.3±3.1 |
| 500 →1250 | 62.6±0.3 | 93.3±0.3 | 54.3±3.8 | 71.3±4.1 | 40.1±17.9 | 74.6±2.3 | **89.8±1.4** | 82.9±3.5 |
| 500 →1500 | 58.0±0.2 | 95.4±0.4 | 55.7±3.1 | 72.0±3.7 | 61.8±20.9 | 73.6±3.0 | **90.8±0.4** | 80.0±6.1 |
| 750 →500 | 52.6±0.1 | 56.3±0.5 | 54.0±0.2 | **56.1±1.1** | 53.1±0.5 | **56.9±0.7** | 53.4±0.8 | **55.7±1.1** |
| 750 →1000 | 86.5±0.1 | 89.2±0.4 | 82.2±0.8 | 83.1±2.6 | **87.9±0.2** | 86.2±0.6 | **87.5±0.6** | **87.7±0.7** |
| 750 →1250 | 87.7±0.0 | 95.4±0.4 | 81.3±2.9 | 88.2±3.1 | 89.5±0.5 | **92.5±0.9** | 92.0±0.4 | 92.3±1.3 |
| 750 →1500 | 87.7±0.0 | 95.4±0.4 | 82.5±2.8 | 88.2±3.1 | 89.5±0.5 | **92.5±0.9** | 91.7±0.4 | 92.8±1.6 |
| 1000 →500 | 51.9±0.0 | 56.3±0.5 | 51.9±0.4 | **55.5±0.9** | 51.9±0.2 | **55.0±0.5** | 53.3±0.4 | 53.2±1.7 |
| 1000 →750 | 77.1±0.1 | 82.2±0.3 | 74.8±0.6 | 77.3±1.1 | 77.7±0.4 | **80.5±0.6** | **80.6±1.1** | **80.7±0.9** |
| 1000 →1250 | 92.6±0.0 | 93.3±0.3 | 90.9±0.5 | 88.7±1.0 | 91.2±0.4 | 92.1±0.4 | 91.7±0.4 | **92.5±0.3** |
| 1000 →1500 | 93.0±0.0 | 95.4±0.4 | 91.0±0.6 | 88.3±2.4 | 91.2±0.3 | **93.5±0.6** | 92.0±0.3 | **93.9±0.9** |
| 1250 →500 | 51.3±0.0 | 56.3±0.5 | 51.5±0.4 | 55.7±0.9 | 52.1±0.5 | 53.8±0.5 | 53.5±0.5 | **56.5±1.4** |
| 1250 →750 | 68.9±0.1 | 82.2±0.3 | 70.2±1.2 | 77.0±1.0 | 77.0±0.7 | **80.1±0.3** | 77.8±0.9 | **80.2±0.7** |
| 1250 →1000 | 87.7±0.1 | 89.2±0.4 | 84.7±0.5 | 82.7±2.4 | **88.6±0.5** | 88.4±0.7 | 87.4±0.7 | **88.5±0.5** |
| 1250 →1500 | 94.1±0.1 | 95.4±0.4 | 93.0±0.3 | 89.5±2.4 | 92.7±0.1 | **94.3±0.4** | **94.0±0.2** | 94.0±0.7 |
| 1500 →500 | 50.8±0.1 | 56.3±0.5 | 50.6±0.5 | **56.2±0.6** | 51.8±0.2 | 53.2±0.5 | 51.8±0.4 | 54.0±2.2 |
| 1500 →750 | 63.3±0.1 | 82.2±0.3 | 64.3±1.2 | 78.2±1.6 | 75.0±0.6 | **80.0±0.2** | 76.4±0.8 | **80.4±1.2** |
| 1500 →1000 | 83.9±0.1 | 89.2±0.4 | 82.8±0.7 | 84.7±1.8 | **88.4±0.4** | 87.4±0.4 | **88.9±0.4** | **88.5±0.8** |
| 1500 →1250 | 92.9±0.1 | 93.3±0.3 | 90.6±0.5 | 88.7±2.3 | **93.0±0.3** | 92.1±0.5 | **92.6±0.2** | 92.4±0.8 |

to reduce the sample dimensionality from 5000 to 5. The NN has two hidden layers of dimensions 10 and 5 (the encoding dimension). For each task, we train the NN as a classifier on the source data with a cross entropy loss. We then cut the output layer and use the trained NN to encode both the source and target data into 5 dimensions. This dimension reduction is the input data of all the algorithms on the benchmark and we trained all the competitors on this data. The results are shown in Table 6.

Table 3: Amazon dataset, unsupervised. Tasks are B: books, D: Dvd, E: Electronics, K: kitchen

| Task | Source | Target | DANN | SA | KMM | OT | SHOT | KRDA |
|---|---|---|---|---|---|---|---|---|
| B → D | 79.9±0.1 | 79.4±0.1 | **79.9±0.1** | **79.9 ± 0.1** | **79.9±0.1** | 79.7±0.1 | 79.7±0.6 | **80.0±0.2** |
| B → E | 69.2±0.2 | 72.7±0.0 | 69.9±0.4 | **73.0 ± 0.1** | 71.8±0.1 | **73.0±0.1** | 72.6±0.7 | **73.0±0.1** |
| B → K | 75.7±0.2 | 76.2±0.1 | 75.8±0.2 | **76.1 ± 0.1** | **76.3±0.1** | 76.2±0.1 | 76.2±0.2 | 76.2±0.1 |
| D → B | 75.1±0.1 | 75.6±0.1 | 75.1±0.2 | 75.3 ± 0.1 | **75.5±0.1** | 75.2±0.1 | 75.5±0.1 | 75.3±0.1 |
| D → E | 71.0±0.3 | 74.6±0.1 | 71.1±0.7 | 74.3 ± 0.1 | 72.7±0.2 | 74.6±0.1 | **73.8±0.9** | **74.4±0.2** |
| D → K | 74.9±0.1 | 77.3±0.1 | 74.8±0.5 | **77.5 ± 0.1** | 76.2±0.1 | **77.5±0.1** | 77.0±0.5 | 76.3±0.2 |
| E → B | 70.7±0.1 | 71.5±0.0 | 70.6±0.4 | 71.2 ± 0.2 | 70.9±0.2 | 71.1±0.1 | **71.3±0.3** | 70.9±0.2 |
| E → D | 72.3±0.1 | 73.6±0.0 | 72.1±0.4 | 73.1 ± 0.2 | **73.3±0.4** | 72.7±0.1 | 72.8±0.3 | 72.6±0.5 |
| E → K | 86.1±0.1 | 86.2±0.1 | 85.5±0.5 | **85.8 ± 0.2** | 83.5±0.3 | **86.0±0.1** | 85.6±0.6 | **85.2±1.0** |
| K → B | 71.2±0.2 | 71.4±0.1 | 71.2±0.3 | 71.6 ± 0.1 | **71.7±0.1** | 71.8±0.0 | 71.6±0.1 | 69.4±0.3 |
| K → D | 70.8±0.1 | 72.8±0.0 | 70.7±1.1 | 72.7 ± 0.3 | 70.6±0.0 | 72.8±0.2 | 70.6±1.0 | **72.8±0.7** |
| K → E | 84.0±0.0 | 84.4±0.0 | 84.0±0.2 | **84.3 ± 0.0** | 84.2±0.0 | **84.3±0.0** | 84.2±0.1 | 84.1±0.1 |

## 7 CONCLUSION

In this paper, we presented a novel transfer learning framework that exploits recent advances in density estimation techniques in order to transfer samples from a source domain to a target domain with a distribution shift using Knothe-Rosenblatt transport. The property that is invariant by the transfer performed by KRDA is the one dimensional conditional quantile distributions from the source in the target domain space, using an autoregressive setup. We showed that KRDA is state of the art on small to moderate dimensional tasks, often outperforming competitors especially in the case where there are few data samples.

Future work includes extending KRDA to the semi-supervised domain adaptation where a few labels are present in the target data. Another interesting research direction is to use other modern density estimators such as normalizing flows or variational autoencoders, in place of autoregressive mixture density nets.

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

# Knothe-Rosenblatt transport for Unsupervised Domain Adaptation

## Supplementary material

## A  EXPERIMENTS

### A.1  INTER-TWINNING MOONS

Table 4: Inter-twinning moons unsupervised

| Task | Source | Target | CDAN | DAN | JAN | TCA | KRDA |
|------|--------|--------|------|-----|-----|-----|------|
| 10° | 100±0 | 100±0 | **100±0** | 50.0±0.1 | **100±0** | **100.0±0** | **100±0** |
| 20° | 99.9±0.1 | 100±0 | 99.8±0.2 | 48.5±3.4 | **100±0** | **100.0±0** | **100±0** |
| 30° | 96.6±0.40 | 100±0 | 97.8±2.1 | 56.1±13.7 | **99.7±0.7** | **100.0±0** | **100±0** |
| 40° | 73.1±2.50 | 100±0 | 80.4±14.4 | 50.0±0.1 | 93.2±7.9 | **98.0±0.8** | **98.4±2.2** |
| 50° | 41.5±2.0 | 100±0 | 47.4±24.5 | 50.0±0.1 | 89.8±4.3 | 80.1±2.0 | **98.4±1.5** |
| 60° | 28.7±0.6 | 100±0 | 30.1±5.0 | 50.0±0.1 | 87.4±4.3 | 44.8±4.7 | **90.5±2.8** |
| 70° | 23.3±0.4 | 100±0 | 24.9±1.7 | 49.9±0 | 62.2±34.9 | 24.9±0.6 | **84.4±0.8** |
| 80° | 20.4±1.7 | 100±0 | 26.7±10.4 | 50.0±0.1 | 49.9±22.6 | 20.5±0.2 | **81.2±3.4** |
| 90° | 18.1±0.3 | 100±0 | 17.2±0.7 | 50.0±0.1 | 27.9±14.1 | 17.8±0.4 | **71.1±3.9** |

### A.2  HEPMASS

Table 5: HEPMASS unsupervised domain adaptation

| solution | Source | Target | CDAN | DAN | JAN | TCA | KRDA |
|----------|--------|--------|------|-----|-----|-----|------|
| 500 →750 | 68.7±0.1 | 82.2±0.3 | 56.0±4.0 | 51.3±0.7 | 58.2±1.9 | 67.2±0.4 | **75.6±2.3** |
| 500 →1000 | 67.2±0.1 | 89.2±0.4 | 49.2±5.6 | 51.4±4.1 | 56.1±3.8 | 71.7±0.4 | **80.3±3.1** |
| 500 →1250 | 62.6±0.3 | 93.3±0.3 | 53.7±8.8 | 48.9±2.7 | 54.9±2.3 | 77.9±0.3 | **82.9±3.5** |
| 500 →1500 | 58.0±0.2 | 95.4±0.4 | 56.3±9.1 | 52.8±0.3 | 54.3±2.6 | 75.9±0.5 | **80.0±6.1** |
| 750 →500 | 52.6±0.1 | 56.3±0.5 | 54.2±0.5 | 50.4±0.8 | **54.9±0.9** | 53.7±0.2 | 55.7±1.1 |
| 750 →1000 | 86.5±0.1 | 89.2±0.4 | 82.6±0.4 | 49.0±0.2 | 81.8±1.7 | 84.5±0.2 | **87.7±0.7** |
| 750 →1250 | 87.7±0.0 | 95.4±0.4 | 82.5±3.1 | 52.1±7.4 | 77.8±3.9 | 85.7±0.1 | **92.3±1.3** |
| 750 →1500 | 87.7±0.0 | 95.4±0.4 | 74.8±7.9 | 56.1±11.8 | 76.6±2.1 | 85.7±0.1 | **92.8±1.6** |
| 1000 →500 | 51.9±0.0 | 56.3±0.5 | **52.0±0.5** | 50.0±0.0 | 51.2±0.6 | **53.1±0.1** | 53.2±1.7 |
| 1000 →750 | 77.1±0.1 | 82.2±0.3 | 74.8±1.2 | 51.5±0.0 | 76.2±1.1 | 73.6±0.2 | **80.7±0.9** |
| 1000 →1250 | 92.6±0.0 | 93.3±0.3 | 91.2±0.5 | 57.2±17.3 | 88.8±0.4 | 91.1±0.1 | **92.5±0.3** |
| 1000 →1500 | 93.0±0.0 | 95.4±0.4 | 90.6±0.6 | 61.4±17.0 | 88.6±1.3 | 91.7±0.1 | **93.9±0.9** |
| 1250 →500 | 51.3±0.0 | 56.3±0.5 | 51.3±0.3 | 50.0±0.0 | 52.1±0.6 | 51.7±0.1 | **56.5±1.4** |
| 1250 →750 | 68.9±0.1 | 82.2±0.3 | 69.8±0.9 | 54.8±12.6 | 76.8±0.7 | 69.8±0.2 | **80.2±0.7** |
| 1250 →1000 | 87.7±0.1 | 89.2±0.4 | 85.2±0.3 | 69.9±19.1 | 85.3±0.7 | 86.5±0.2 | **88.5±0.5** |
| 1250 →1500 | 94.1±0.1 | 95.4±0.4 | 93.2±0.4 | 72.6±23.8 | 93.3±0.7 | 93.0±0.1 | **94.0±0.7** |
| 1500 →500 | 50.8±0.1 | 56.3±0.5 | 50.2±0.4 | 50.0±0.2 | 51.3±0.6 | 51.3±0.1 | **54.0±2.2** |
| 1500 →750 | 63.3±0.1 | 82.2±0.3 | 62.1±0.9 | 53.1±10.3 | 73.9±0.6 | 66.0±0.2 | **80.4±1.2** |
| 1500 →1000 | 83.9±0.1 | 89.2±0.4 | 82.3±1.4 | 75.8±11.5 | 85.9±1.7 | 83.4±0.2 | **88.5±0.8** |
| 1500 →1250 | 92.9±0.1 | 93.3±0.3 | **91.0±0.6** | 84.1±18.8 | 91.3±0.2 | **91.9±0.3** | 92.4±0.8 |

### A.3  AMAZON

Table 6: Amazon dataset, unsupervised. Tasks are B: books, D: Dvd, E: Electronics, K: kitchen

| solution | Source | Target | CDAN | DAN | JAN | TCA | KRDA |
|---|---|---|---|---|---|---|---|
| B → D | 79.9±0.1 | 79.4±0.1 | **79.7±0.6** | 63.1±15.5 | 59.5±13.5 | **79.9±0.1** | **80.0±0.2** |
| B → E | 69.2±0.2 | 72.7±0.0 | 64.5±8.5 | 59.6±10.8 | 70.9±0.4 | 69.9±0.2 | **73.0±0.1** |
| B → K | 75.7±0.2 | 76.2±0.1 | 75.8±0.1 | 55.2±11.6 | 70.9±11.5 | 75.8±0.1 | **76.2±0.1** |
| D → B | 75.1±0.1 | 75.6±0.1 | **75.3±0.2** | 69.3±9.1 | 70.2±10.9 | **75.0±0.0** | **75.3±0.1** |
| D → E | 71.0±0.3 | 74.6±0.1 | 66.9±9.3 | 68.0±9.9 | 67.0±9.9 | 71.2±0.2 | **74.4±0.2** |
| D → K | 74.9±0.1 | 77.3±0.1 | 75.1±0.2 | 60.2±14.0 | 62.1±13.0 | 75.3±0.0 | **76.3±0.2** |
| E → B | 70.7±0.1 | 71.5±0.0 | **70.7±0.4** | 58.3±11.7 | 58.6±11.5 | **71.0±0.1** | 70.9±0.2 |
| E → D | 72.3±0.1 | 73.6±0.0 | **71.4±1.3** | 66.5±9.8 | 63.8±12.6 | **72.1±0.1** | **72.6±0.5** |
| E → K | 86.1±0.1 | 86.2±0.1 | **85.5±0.4** | 83.5±5.4 | 64.4±19.6 | **86.0±0.2** | 85.2±1.0 |
| K → B | 71.2±0.2 | 71.4±0.1 | **71.2±0.2** | 69.7±3.6 | 62.6±11.6 | **71.2±0.1** | 69.4±0.3 |
| K → D | 70.8±0.1 | 72.8±0.0 | 70.5±0.4 | 62.4±11.7 | 71.0±0.2 | 70.7±0.0 | **72.8±0.7** |
| K → E | 84.0±0.0 | 84.4±0.0 | **84.0±0.2** | **84.1±0.2** | 77.2±15.4 | 83.9±0.1 | **84.1±0.1** |

