# OpenReview forum: "Knothe-Rosenblatt transport for Unsupervised Domain Adaptation"
_ICLR.cc/2022/Conference — ICLR 2022 Submitted_

### Official Review · Reviewer_HHVi · 2021-10-26

**Correctness:** 3
**Technical Novelty And Significance:** 2
**Empirical Novelty And Significance:** 2
**Recommendation:** 3
**Confidence:** 4

**Main Review:**

**Strengths**:
- The paper is the first work that applies Knothe-Rosenblatt to the unsupervised domain adaptation problem. So, originality is an advantage of the paper.
- The paper is well-written, easy to follow. Parts of the paper are well organized.
- The author cleverly exploits the nature of Knothe-Rosenblatt that is well-fitted to the density estimation tool like an autoregressive model.
- Literature is cited quite sufficiently.
**Weaknesses**:
I think that the paper can become a much stronger paper if the author can address the following weaknesses:
- The limitation of autoregressive models is also the limitation of KRDA as mentioned by the authors,  so the KRDA is not scalable in terms of the number of dimensions. Hence, I suggest that the authors should propose an additional "deep" KRDA method that uses a neural net as a dimensional reduction method and feature extraction method (as authors did on Amazon datasets with the difference that the neural net is also trained with an adaptation loss ), and then apply KRDA on the latent space. Optimal transport losses are mostly utilized on this approach [1] - OT, [2] - unbalanced OT, [3] - partial OT. Also, it is necessary to compare deep KRDA with these models on some famous datasets such as SVHN, MNIST, UPS, Office-home, etc. For me, I consider KRDA is a deep domain adaptation method since it uses RNADE. So, comparison with more deep DA methods is needed.

[1] DeepJDOT: Deep Joint Distribution Optimal Transport for Unsupervised Domain Adaptation

[2] Unbalanced minibatch Optimal Transport; applications to Domain Adaptation

[3] Improving Mini-batch Optimal Transport via Partial Transportation

- I am not sure about the benefit of KRDA over the OT-based DA. For example, JDOT does not need to smooth the source and the target density, namely, JDOT only needs to deal with empirical measures.  Since the performance on the Amazon dataset of KRDA is not superior to JDOT, I suggest the authors should do a speed comparison between methods.  I believe that training RNADE is also expensive.

- It seems that KRDA needs invertible CDFs for the conditional distributions of RNADE (e.g. Mixture of Gaussians in the paper). I wonder if the authors can discuss families of distributions that can be utilized here and compare their performance. Also, the authors should motivate the usage of mixtures of Gaussians here.

- The authors have not published the code yet. So, I could not run any quick verification of the experiments. I encourage the author to publish the code in the discussion phase.

**Comments and Questions:**
- Could the authors make a discussion between Knothe-Rosenblatt transportation and sliced optimal transport. I know that the difference is that sliced OT is based on random projections while Knothe-Rosenblatt is based on conditional distributions. However, I think a discussion is a good contribution here.
- RNADE is trained with the maximum likelihood estimator (KL divergence). Could Knothe-Rosenblatt transportation be transformed into a notation of divergence between the source measure and the target measure?

**Summary Of The Paper:**

The paper proposes Knothe-Rosenblatt Domain Adaptation (KRDA) for unsupervised domain adaptation with covariate shift. The key idea of the paper is to use an autoregressive model (RNADE) to estimate the density of source distribution and target distribution, then transport the estimated source density to the estimated target density by Knothe-Rosenblatt transportation. In particular, the transportation maps are constructed on each conditional density that is obtained from fitted autoregressive models. The authors evaluated the proposed model on synthetic data (mixture of Gaussians, Inter-twinning moons), and real data (Hepmass, Amazon dataset).

**Summary Of The Review:**

I consider the paper is a methodology paper, so I believe that more experiments are needed e.g. comparison with deep DA methods including OT-based deep DA, computational speed comparison between these methods. Also, some discussion is needed as mentioned in the review part.

---

### Official Review · Reviewer_kupY · 2021-11-01

**Correctness:** 3
**Technical Novelty And Significance:** 2
**Empirical Novelty And Significance:** Not applicable
**Recommendation:** 3
**Confidence:** 4

**Main Review:**

Strength:

(1)	A new KRDA method is proposed.

(2)	Both synthetic and real-world experiments are done.

The overall idea is estimating the source and target distributions and then learn the matching through transport. The idea is simple and not new. Here are some detailed comments:

(1)	 The related works are out-of-date, especially for subspace based methods. Please consider the more recent works, e.g., [ref1] and [ref2].

[ref1] Visual Domain Adaptation with Manifold Embedded Distribution Alignment
[ref2] Subdomain adaptation with manifolds discrepancy alignment

(2)	Only using the last linear layer to capture the domain difference is some weak. You may consider the structure in DAN [ref3] where the first several layers are generic and thus fixed, the medium layers are fine-tuned, and the last several layers are leaned to be aligned.

[ref3] Learning Transferable Features with Deep Adaptation Networks

(3)	The superiority of KRDA on small data with small dimension is highlighted. In this sense, it is more preferred to compare with those subspace transfer methods. However, the current baselines are not state-of-the-art, which makes the empirical results less convincing. Moreover, some benchmark datasets, e.g., office 31 which has small data size, are expected to be tested.


**Summary Of The Paper:**

In this paper, the authors propose a Knothe-Rosenblatt Domain Adaptation (KRDA) method. Specifically, it exploits autoregressive model using a mixture of Gaussians to accurately model the different sources, and then takes advantage of the triangularity of the autoregressive models to build an explicit mapping of the source samples into the target domain. Experiments on both synthetic and real-world datasets are done to verify KRDA.

**Summary Of The Review:**

See comments above.

---

### Official Review · Reviewer_paqc · 2021-11-02

**Correctness:** 2
**Technical Novelty And Significance:** 2
**Empirical Novelty And Significance:** 1
**Recommendation:** 3
**Confidence:** 3

**Main Review:**

+ Strengths:

The usage of density estimation by autoregressive model for Knothe-Rosenblatt transport for unsupervised domain adaptive obtains good performances.

+ Weakness:

It seems that the contribution is quite incremental: (i) using optimal transport to transfer source to target for domain adaptation is not new (ii) Although the usage of density estimation by autoregressive model for Knothe-Rosenblatt transport is novel, its motivation is unclear -- e.g., about the advantage of the proposed approach compared with other optimal transport-based approaches for domain adaptation.

+ Detailed comments:

1. What are the motivations for the proposed method (e.g., using density estimation by autoregressive model for Knothe-Rosenblatt transport) compared with other optimal transport-based approaches for domain adaptation? (e.g., standard optimal transport, or sliced-optimal transport for its fast computation and its variants)

2. The choice of autoregressive modeling for density estimation is not well-motivated. What's kind of applications suitable for this density estimation? (In which cases, this approach may not be a good choice). I think a discussion about the choice of autogressive modeling for density estimation will help readers.

3. It seems that the proposed approach is closely related with the sliced-Wasserstein. I think it is a good baseline for the proposed approach (simple + fast computation) -- autogressive modeling density estimation is also based on 1d-density estimation.

4. For experiments, why does the proposed approach obtain better results than the OT approach? The authors should discuss the experimental results in detail.

5. For experiments, the setting may be limited when only binary classification is considered for both source and target. As illustrated in Figurer 2, when the source and target have different numbers of classes, is it possible to use the proposed approach?





**Summary Of The Paper:**

The authors consider the unsupervised domain adaptation (UDA) by using the Knothe-Rosenblatt transport. The authors propose to estimate the density of both source and target data via autoregressive modeling (i.e., decompose the d-dimensional density into d one-dimensional conditional densities), then apply Knothe-Rosenblatt transport to obtain the transfer map from the source to target for UDA.




**Summary Of The Review:**

It seems that the contribution is quite incremental. The usage of optimal transport-based approaches for domain adaptation is not new. The proposed OT-based approach is not well-motivated, its advantage is not cleared yet (although it has better performances, there are no discussions why such results are expected)

---

### Official Review · Reviewer_KT59 · 2021-11-02

**Correctness:** 3
**Technical Novelty And Significance:** 2
**Empirical Novelty And Significance:** 2
**Recommendation:** 3
**Confidence:** 4

**Main Review:**

Strengths
As far as I can check, one of the main originality of the paper is to use an auto-regressive model to estimate the conditional marginals allowing a relatively easy application of the Knothe-Rosenblatt transport. Another point is the application to a dataset in high-energy physics with interesting results.

Weaknesses
The idea of using optimal tranport based approaches is not really novel and many approaches exist. This paper proposes a novel strategy but there is no real strong point that makes this contribution significantly different from others, in particular with respect to other optimal transport strategies.
The contribution is mainly algorithmic, there is no theoretical results supporting the method (convergence properties, generalization bounds, ...).
The experimental evaluation uses mainly existing competitors which is good, but since the method is in the line of optimal transport (OT) strategies. A deeper comparison with other OT baselines for domain adaptation (different regularization or other strategies: JDOT [Courty et al., NeurIPS 2017], COOT [Vayer et al., NeurIPS 2020], ...) would reinforce the interest of the approach and possibly exhibits interesting properties. The expressiveness with respect to other OT methods should be developped and better compared.
I do not find the experimental setup sufficiently clear, a detailed paragraph (in supplementary material). It does not seem that the paper tunes the competitors according to their classic usage in their original paper, the architectures seem limited both in terms of number of layers and in terms of the number of neurons per layer, this is not clear that the experimental setting is fair for the other competitors.
There is no benchmarks in vision, which limits the comparison with existing results in the State of the Art.
The authors claim that their approach is better for small datasets, why not, but this should be better analyzed: using more datasets (going to the few-shot learning setting?), tune other competitors adequately for the considered problems, maybe deep learning solutions are not the best here (even though, they should be considered as well of course.
Running time indicators should be useful to justify gains in terms of computational efficiency.


**Summary Of The Paper:**

This paper proposes to tackle unsupervised domain adaptation by following a classic trend aiming at finding an alignment between source and target domains. More specifically, the method uses a Knothe-Rosenblatt transport approach which applies a one-dimensional optimal transport to all conditional marginals of one distribution into another. The method uses an autoregressive density estimation from the RNADE algorithm based on a Gaussian Mixture strategy. The experimental evaluation compares the approach and other classic domain adaptation benchmarks on synthetic data, on the Amazon benchmark and a benchmark in higgh-energy physics.


**Summary Of The Review:**

The paper presents a novel idea consisting in using an auto-regressive approach to estimate the condition marginal densities which allows the to use a more efficient application of the Knothe-Rosenblatt optimal transport (OT). This is new and interesting.
However, the comparison both in terms expressiveness and experiments with respect to other OT is rather limited and should be extended.
The experimental setup is not completely clear and thus not clearly convincing.
Some benchmarks in vision are not considered.
There is no theoretical results supporting the paper.

Overall, there is the beginning of an interesting idea but that is not enough developed for ICLR.

---

### Decision · Program_Chairs · 2022-01-20

**Decision:**

Reject

**Comment:**

This paper proposes to address the problem of domain adaption using Knothe-Rosenblatt transport withe the method denoted as KRDA . The main idea is to perform density estimation of the different distributions with mixture of Gaussians and then estimate a  an explicit mapping between the distribution using  Knothe-Rosenblatt. Experiments show that the proposed method works well on toy and real life datasets.

 The paper had low score during the reviews (3,3,3,3). While the reviewers appreciated the idea, they felt that the originality of the method is not well justified compared to a number of existing UDA approaches using OT. Also the reviewers noted several important references missing and that should also be compared during the numerical experiments. A discussion about the limits of the method in high dimension would also be very interesting.

The authors did not provide a reply to the reviewers' comments so their opinion stayed t same during the discussion. The paper is then rejected and the AC strongly suggests that the authors take into account the numerous comments from the reviewers before re-submitting ton a new venue.